# Evaluation of Fiber-Reinforced Modular Soft Actuators for Individualized Soft Rehabilitation Gloves

**Shota Kokubu** [1] , **Yuanyuan Wang** [1] , **Pablo E. Tortós Vinocour** [1], **Yuxi Lu** [1], **Shaoying Huang** [2] , **Reiji Nishimura** [3], **Ya-Hsin Hsueh** [4] **and Wenwei Yu** [1,5,*]

1   Department of Medical Engineering, Graduate School of Engineering, Chiba University, Chiba 263-8522, Japan; s.kokubu@chiba-u.jp (S.K.); wangyuanyuan@chiba-u.jp (Y.W.); ccta0833@chiba-u.jp (P.E.T.V.); yuxi.lu@chiba-u.jp (Y.L.)
2   Engineering Product Development, Singapore University of Technology and Design, Singapore 487372, Singapore; huangshaoying@sutd.edu.sg
3   Department of Plastic and Reconstructive Surgery, Jikei University School of Medicine, Tokyo 105-8461, Japan; nishimura@jikei.ac.jp
4   Department of Electronic Engineering, National Yunlin University of Science and Technology, Yunlin 64002, Taiwan; hsuehyh@yuntech.edu.tw
5   Center for Frontier Medical Engineering, Chiba University, Chiba 263-8522, Japan
*   Correspondence: yuwill@faculty.chiba-u.jp; Tel.: +81-43-290-3231

**Abstract:** Applying soft actuators to hand motion assist for rehabilitation has been receiving increasing interest in recent years. Pioneering research efforts have shown the feasibility of soft rehabilitation gloves (SRGs). However, one important and practical issue, the effects of users' individual differences in finger size and joint stiffness on both bending performance (e.g., Range of motion (ROM) and torque) and the mechanical loads applied to finger joints when the actuators are placed on a patient's hand, has not been well investigated. Moreover, the design considerations of SRGs for individual users, considering individual differences, have not been addressed. These, along with the inherent safety of soft actuators, should be investigated carefully before the practical use of SRGs. This work aimed to clarify the effects of individual differences on the actuator's performance through a series of experiments using dummy fingers designed with individualized parameters. Two types of fiber-reinforced soft actuators, the modular type for assisting each joint and conventional (whole-finger assist) type, were designed and compared. It was found that the modular soft actuators respond better to individual differences set in the experiment and exhibit a superior performance to the conventional ones. By suitable connectors and air pressure, the modular soft actuators could cope with the individual differences with minimal effort. The effects of the individualized parameters are discussed, and design considerations are extracted and summarized. This study will play an important role in pushing forward the SRGs to real rehabilitation practice.

**Keywords:** soft robotics; hand rehabilitation; hand finger motion assist; soft actuators; flexion; passive extension; individualization

## 1. Introduction

Worldwide, 80 million people have survived a stroke as of 2016 [1]. At least 65 % of them suffer from hand disabilities [2]. Since hand functions play an important role in maintaining physical performance in daily living and the quality of life [3], the rehabilitation of hand functions is critical and always in urgent demand, impelling the research and development of hand function rehabilitative devices.

Hand rehabilitation devices have been developed based on different types of rigid mechanics such as linkage-slide mechanisms [4], cable-actuated extension, spring-return flexion mechanisms [5], and slider-crank-like mechanisms [6]. However, the hand rehabilitation devices based on rigid mechanics are less flexible in movement, heavy to carry, complicated to fabricate, and have safety concerns. These problems have been recently

overcome by soft actuators in hand motion assistive gloves [7–10]. The most current soft actuators for rehabilitation and power-assist glove systems, consisting of molded elastomeric chambers with fiber reinforcements that induce specific bending, twisting, and extending trajectories under fluid pressurization, control only one axis of motion for each joint [7–9], except multi-pocket soft actuators supporting metacarpophalangeal (MCP) joint abduction–adduction [10].

Research efforts have been further made to validate the concept of soft rehabilitation gloves (SRGs) with different assessment levels and to show their feasibility. In [11], the capability of a glove called AirExGlove for patients affected by post-stroke muscle spasticity was reported. With the help of the glove, the patients affected by clenched fist deformity could perform any intermediate hand-opening movement between whole hand opening and closure. In another work, both the flexion and extension torque of SRGs on the MCP joint of a dummy finger of both a healthy participant and a stroke patient were tested [12]. Furthermore, Yap et al. tested the Range of motion (ROM) and grip force of SRGs on five healthy participants and a stroke patient, and their results showed the validity of the SRGs for supporting ADLs [13]. In [14], the active ROM of prototype SRGs with a Kapandji test and a standardized Box-and-Block test was tested on a patient with muscular dystrophy, which demonstrated an increase in the precision and accuracy of the grasp when wearing the glove. There have been reports of SRGs for the complex motor function of the thumb. Shiota et al. proposed a soft actuator for thumb movement support and reported the measured ROM and the maximum contact force of the thumb finger in an extended Kapandji test [15].

For most of the SRGs reported in the literature, the user's individual differences, such as hand finger size (length dimension), ROM of finger joints, and joint stiffness, are not considered. However, their effects could affect the performance and effectiveness of rehabilitation. The customization of SRGs is the next critical step towards establishing the SRGs as an ordinary assistive and rehabilitation tool. Although it has been claimed that some SRGs and even products could deal with the individual differences [11,16], fewer efforts have been made to clarify the effects of these individual differences on the assistive and rehabilitation performance and joint loading, which are not trivial. As was shown in studies of hand orthopedics and surgery, the ROM of finger joints is not constant, even in the healthy group [17,18]. Especially, the joints with smaller ROM may not be able to disperse external forces, thus having a higher risk of injuries to their joint support tissues [18]. Additionally, when applying loads to finger joints through orthosis, the improper application of force may injure the skin, ligaments, and cartilage. Furthermore, if the applied force is not perpendicular to the joint's axis of rotation, there is a risk that the collateral ligament may be unevenly stressed, causing joint instability, or the force is exerted to the articulating surface, which damages articular cartilage [18]. For all these reasons, an orthosis needs to be adapted to individual anatomical variations, structural abnormalities, defects, and changes in the pathological condition [19].

Although there are few reports on the complications caused by robot-assisted rehabilitation [20], unlike in strictly controlled clinical trials, in clinical practice, the loosening of orthosis and individual differences can easily cause misalignment, which increases the risk of interruption of rehabilitation, tissue damages, dislocations, and cartilage injury [21]. In addition, abnormal pressure distribution and excessive loads on the articular surface pose the risk of causing osteoarthritis [22]. In particular, during rehabilitation, less apparent deformation and/or injuries of soft tissues around joints, including cartilage, may not be felt by patients with sensory impairment [19]. However, even if the patient's negative impacts, such as the above-mentioned excessive loads to joints, are not prominent each time, during the repetitive training of any rehabilitation procedure, they accumulate. This can be counterproductive and even dangerous for patients with a certain level of hand function impairment. Therefore, it is necessary to carefully investigate the effects of the individual differences on rehabilitation performance regardless of the inherent safety of soft actuators and explore the possible solutions to mitigate the negative effects.

In the research area of soft robotics for SRGs, individual differences have begun to be identified as an important design factor [12,13,15]. In fact, it was reported that misalignment between SRG and hand finger size causes distortion of the natural movement of the hands and discomfort of the wearer after continuous use [14]. However, there has been no clear evidence and systematic investigation on the topic. On the other hand, it is known that to design an orthosis for fingers with movement disorders, it is necessary to decide which joints to move and which joints to fixate [19]. When there is a movement disorder in only one finger joint, if force is applied to the whole finger, there is little effect on the joint with the disorder since the normal joints absorb the assisting force.

Regarding the design consideration of SRGs for individual users, a design interface was developed which allows the remote processing of hand images and a prompt determination of the required length for each soft actuator [14]. In [23], the requirements for a hand rehabilitation robotic device were investigated and summarized using a method called quality function development for both patients and medical workers.

To the best of our knowledge, the only assistive glove that has the potential for low-cost customization and repair is the Exo-Glove PM [16], in which fabric-based actuator modules are used with spacers specially designed to absorb individual differences in hand finger size. However, in [16], neither the effects of the size differences have been made clear, nor have the design considerations for the actuator modules and spacers been clarified. Moreover, the pathological conditions of the fingers, including the ROM and joint stiffness of fingers, have not been addressed.

This study aimed to evaluate the basic design of a new fiber-reinforced modular soft actuator for individualized SRGs. The modular soft actuators were compared with the conventional (whole-finger assist) actuators in the performance evaluation. Through a series of experiments, we also aimed to clarify the effects of two aspects of individual differences, the hand finger size and the joint stiffness.

In order to ensure the objectivity of the evaluation of the actuators for the SRGs, the experiments used dummy fingers designed with typical individualized parameters instead of healthy subjects and patients.

This paper is organized as follows: First, the requirements for individualization are presented, which is followed by the introduction of the two types of soft actuators under study, the modular and the conventional one, based on the requirements for individualization. Section 4 presents the fabrication of actuators and dummy fingers, especially size and stiffness, considering individual differences and typical pathological conditions. After Sections 5 and 6, the evaluation method and results of the actuator's performance considering individual differences are shown. Finally, we present the potential and contribution of individualization in the proposed fiber-reinforced modular soft actuators and conclude this work.

## 2. Individual Differences of Fingers and Requirements for Individualized SRGs

This section is dedicated to the analysis of the individual variations of hand finger's size, the difference in ROM and stiffness caused by pathological conditions, as well as the explanation of the requirements of actuators for individualized SRGs.

### 2.1. Hand Finger's Size

When considering individual differences in hands, the first factor to consider is the difference in finger size, especially length dimensions. It is clear that there are individual differences in phalangeal and joint size depending on the country, age, and gender. This is also true for patients with hand injuries who need rehabilitation. Therefore, finger size is the most basic and essential factor to consider in individualized SRGs. The Human Life Engineering Research Center investigated the finger dimensions of Japanese people and showed that the average length from the proximal end of metacarpal bones to the tip of the fingers is 185.6 ± 7.9 mm (mean ± s.d.) for men and 171.9 ± 7.7 mm (mean ± s.d.) for women [24].

Differences in the finger size lead to changes not only in the overall length of the fingers but also in the relative position of each joint. If the importance for soft actuators to deal with the individual difference was further made clear, soft actuators should be designed to adapt to size differences, with minimal effort. In this work, we focused on the finger dimension variations of Japanese people. However, the same approach could be applied to other nations.

### 2.2. Joint ROM and Stiffness

The other individual differences that should be considered are the change in joint ROM and stiffness caused by pathological factors. Restricted joint ROM and increased joint stiffness are well observed in the hands of paralyzed patients. Additionally, their degree varies depending on the individual's pathological condition and needs to be fully considered when performing rehabilitation. In motor paralysis induced by stroke, which is the focus of this work, such joint contractures often occur due to the non-learning use of the paralyzed hand [25]. This is because the muscles and soft tissues become stiff due to the imbalance of the antagonist muscles and the maintenance of the paralyzed fingers' flexion. Therefore, these individual differences can change depending on the time from the occurrence of motor paralysis and the frequency of use of the paralyzed hand. In [26], Kamper et al. evaluated the flexion torque generated in contracted fingers due to motor paralysis and reported that the total flexion torque of four fingers (excluding the thumb) is 0.5–4 Nm, depending on the degree of individual symptoms. In another paper [27], Kawasaki et al. also reported that it is necessary to have a bend assist torque of 0.1–0.2 Nm at each joint to perform effective rehabilitation for patients with contracture, from the perspective of physiotherapists. If the assist power of the hand rehabilitation device for joint stiffness is small, the patient may not be able to move the fingers sufficiently, and there is a concern that the rehabilitation effect will be significantly compromised. Therefore, soft actuators for individualized SRGs need to be designed to adapt to different joint ROM and stiffness, and it is important to not compromise their support performance.

### 2.3. Requirements of SRGs Actuators for Individual Adaptation

Based on the analysis above, to push forward the SRGs to real rehabilitation practice, the following are the requirements for an SRG actuator design for adaptions of individual differences:

- A mechanical design that can adapt to differences in finger dimensions;
- Sufficient assist force for coping with joint stiffness;
- Customizability to different individual joint levels;
- Low cost of fabrication, customization, and assembly for individualization.

## 3. Actuator Design

This section contains a detailed introduction of the design of two types of soft actuators, a modular type designed to adapt to individual differences and a conventional (whole-finger assist) type.

### 3.1. Modular Type

There have been several types of modular soft actuators for hand finger rehabilitation for the purpose of reducing both the cost and time required for the fast prototyping of individualized soft gloves. This is because right after the prescription of an SRG, for modular soft actuators, rather than prototyping the whole soft fingers, it is only necessary to print the rigid connectors and assemble them with ready-made, standardized modular soft actuators for adapting the glove to a patient's individual characteristics. On this topic, Hu et al. proposed a bellows-type actuator that can be installed for each joint separately and assist bending and extension [28]. Yun et al. also proposed a fabric-based actuator module, demonstrating the possibility of personal customization [16]. In addition, in one of our previous studies [29], an approach to modularize fiber-reinforced elastomer type

actuators for individualization was proposed. Although all of them show the possibility to individualize the soft glove, the fiber-reinforced elastomer type shows a higher bending performance at lower input air pressures among them. In this work, on top of our previous modularizing approach, a new fiber-reinforced modular soft actuator was proposed for investigating the effects of individualization.

The fiber-reinforced modular soft actuator was designed following our previous studies [15,29,30], based on the requirements described in Section 2, while keeping the advantages of traditional fiber-reinforced soft actuators. Unlike conventional ones, it was scaled to a single joint size instead of covering the whole finger. The components of the actuator are a silicone body, a semicircular chamber, a silicone tube, reinforcing fiber (cotton), and 3D-printed rigid connectors. The internal structure of the actuator consists of a tube and a chamber which is semicircular [30] (see the top in Figure 1). The guide groove for the reinforcing fiber was set at a distance of 2 mm on the actuator body surface. A method called hitching was used as the fiber reinforcement in the actuator [30]. This limits the radial expansion and strengthens the bending motion of the actuator. The modular soft actuators were connected by rigid connectors that are glued to both ends of each actuator and connect each module and a rigid connector for setting the proper distance between the joints.

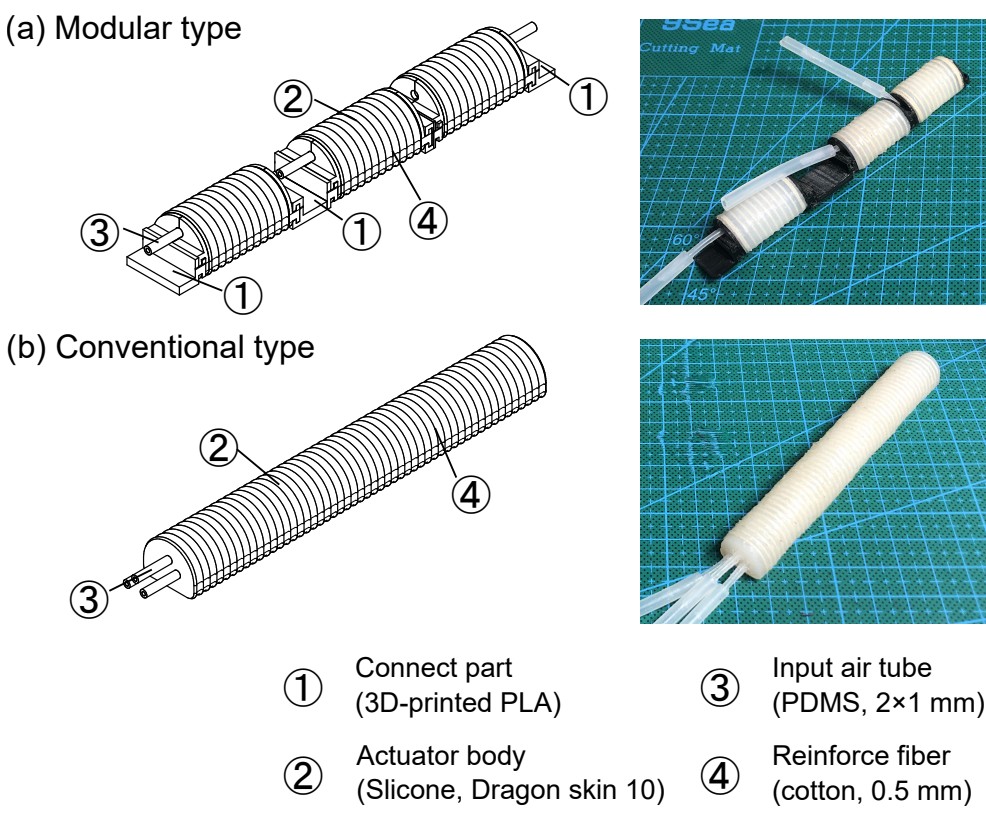

**Figure 1.** Overviews of each type soft actuator. (**a**) Modular type; (**b**) conventional type.

The lengths of the rigid connectors ($L_1$, $L_2$) were calculated by Equations (1) and (2), where $L_{PP}$, $L_{MP}$, $L_{DP}$ are the length of proximal phalanx (PP), medial phalanx (MP), and distal phalanx (DP), respectively. These lengths were acquired from the aforementioned database, based on the average length of the Japanese index finger and its joints [24]. The lengths of the modular actuator and its chamber were 23 mm and 20 mm, respectively. These were determined through trial and error to match the average length of fingers. Regarding the alignment of the finger joint and its supporting actuator, each actuator's center of the chamber needs to be matched with the center of the finger joint, which is critical in terms of finger joint movement support and inherent safety. The other dimensions

of the modular soft actuator were also designed based on overall finger length, width, and distance between joints (Figure 2), with reference to the aforementioned index finger size database [24].

$$L_1 = L_{PP} - 23.0 \ (\text{mm}) \tag{1}$$

$$L_2 = L_{MP} - 23.0 \ (\text{mm}) \tag{2}$$

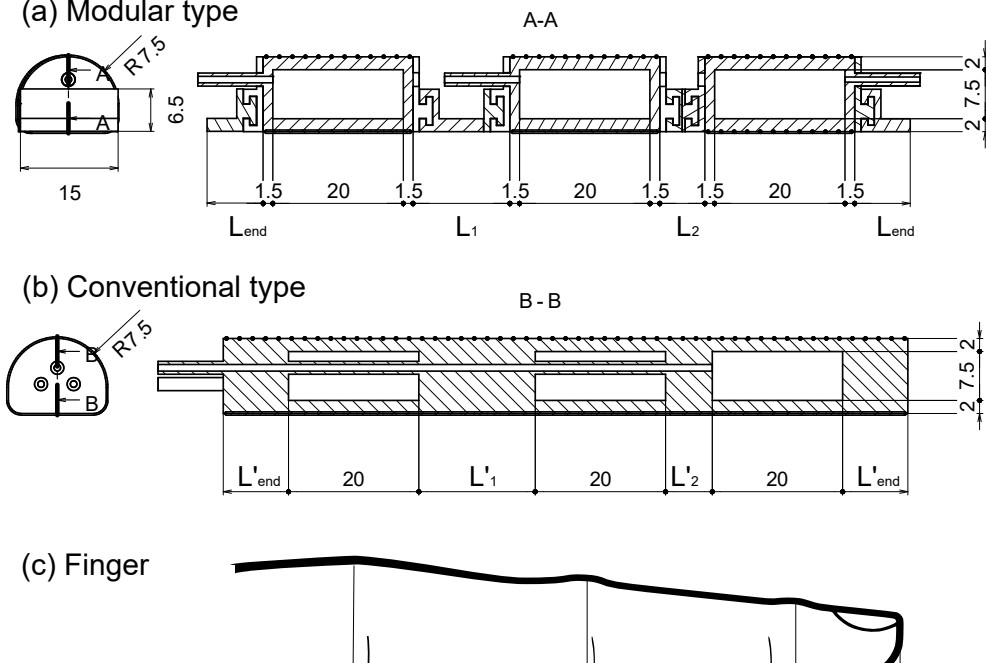

**Figure 2.** Cross sections and dimensions of the each type of soft actuator. (**a**) Modular type; (**b**) conventional type; (**c**) finger.

### 3.2. Conventional Type

As mentioned in Section 1, most current SRGs use single-chamber, conventional soft actuators. In this paper, we chose the fiber-reinforced soft actuator tested and reported in several papers for comparison with the proposed modular type (see the bottom in Figure 1) [12–14]. Since the design consideration and prototyping process have been described, please refer to the paper [30] for the details.

However, to keep the comparison fair, the basic design of the conventional soft actuator was retained, while the chamber size and position were adjusted accordingly with the modular one. In addition, since a single actuator is generally not sufficient to support a change in hand size, the following equation was used to determine the dimension of the conventional soft actuator so that it can be redesigned and fabricated to fit different finger sizes (Equations (3) and (4) and Figure 2).

$$L_1' = L_{PP} - 20.0 \ (\text{mm}) \tag{3}$$

$$L_2' = L_{MP} - 20.0 \ (\text{mm}) \tag{4}$$

### 4. Fabrication

This section presents the parameters used to determine the dimensions of the proposed actuators and dummy fingers for fabrication. Following the introduction of the parameters, the details of the manufacture of the actuators and the dummy fingers are presented. During the evaluation experiments, the fabricated dummy fingers imitate the human fingers with individual differences on each dummy finger.

### 4.1. Parameters of the Individual Differences

The individual differences of fingers in this study, hand finger sizes, joint ROM, and joint stiffness, are reflected by the parameters that describe the dimensions of actuators and the dummy fingers. These parameters are labeled in Figure 2 and summarized in Table 1.

**Table 1.** Parameters of the individual differences.

| | Length | | | Width | Height |
|---|---|---|---|---|---|
| | **PP** | **IP** | **DP** | | |
| Small size (mm) | 35.11 | 23.97 | 17.66 | 14.69 | 11.91 |
| Medium size (mm) | 37.84 | 27.10 | 18.08 | 16.11 | 12.92 |
| Large size (mm) | 42.99 | 30.67 | 21.81 | 18.82 | 14.30 |
| | **DIP joint** | | **PIP joint** | | **MCP joint** |
| Standard ROM (deg) | 80 | | 100 | | 90 |
| Low stiffness (Nmm·deg) | N/A | | N/A | | 0.12 |
| Mid stiffness (Nmm·deg) | N/A | | N/A | | 0.19 |
| High stiffness (Nmm·deg) | N/A | | N/A | | 0.29 |

This work will discuss individual differences by generalizing to the most common index finger size taking into account all fingers, not including the thumb. According to Japan Hand Dimensions Database 2010, hand finger sizes can be represented by three models, small, medium (standard), and large, at a particular age [24]. These models were used to guide the designs of actuators and dummy fingers and used as indicators of the evaluation experiments for the effects of individualization.

The joint ROM was determined with reference to the values of healthy subjects [31]. These values were the ROMs required for finger movement assist and are set as the maximum bending angle of the dummy finger. Regarding the joint stiffness, the pathological situation was limited to the hyperextension of the MP joint, which is often seen in stroke paralyzed patients [25]. The range of stiffness was determined as a torsion spring with a spring constant of three stages, low, middle, and high, with reference to previous research conducted on the MCP joint torque of the index finger [32].

### 4.2. Actuators

Each type of soft actuator was made to fit the large, medium and small sizes based on Equations (1)–(4) and the individual parameters listed in Table 1. It is possible to deal with different finger sizes in the modular soft actuators by changing only the connector without changing the actuator itself. Therefore, only one size of modular soft actuator was made for distal interphalangeal (DIP), proximal interphalangeal (PIP), and metacarpophalangeal (MCP) joints, as well as at three different sizes of rigid connectors. For a fair comparison, three sizes of conventional soft actuators, large, medium, and small, were made.

For the fabrication, first, a mold for molding the silicon body was made with a 3D printer. Silicon (Dragon Skin 10, SMOOTH-ON) was poured into the mold, dried, and cured. A tube was glued to a hole in the silicon body made in advance after molding. Next, a cotton thread with a diameter of 0.5 mm was wound as a reinforcing fiber along the groove on the outer circumference of the silicon body by using the hitching method previously shown on [30]. Furthermore, in the modular soft actuator, we glued the 3D-printed connector to both ends of the silicon body. Finally, a thin silicon layer was applied to the entire surface to fix the reinforcement fibers, and the layer was dried. The molds and connectors used in this study were all made of polylactic acid (PLA) resin.

### 4.3. Dummy Finger

Physical dummy fingers of different sizes were designed and used to objectively evaluate the effectiveness of actuators, SRGs for hand function assist and rehabilitation, as well as exoskeletons [30,33]. Figures 3 and 4, respectively, show the 3D and 2D CAD drawings of the dummy fingers designed for this study which imitate the index finger. The dimensions were determined according to Table 1. All parts were 3D printed from polylactic acid (PLA) for fabrication. For reproducing the effects of joint stiffness, torsion springs were installed at each joint of the dummy finger, as shown in Figure 3. To adjust the joint stiffness, high, mid, and low levels of spring constants were selected according to Table 1.

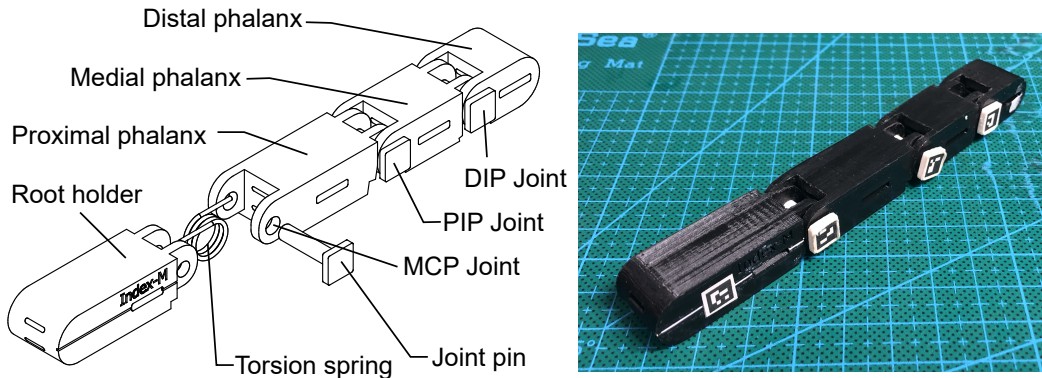

**Figure 3.** Overview of the dummy finger.

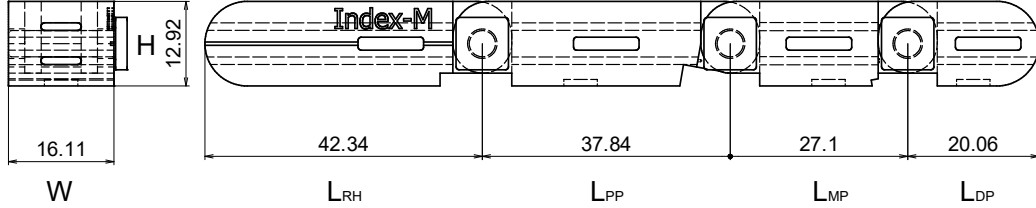

**Figure 4.** Dimensions of the dummy finger.

## 5. Evaluation

To evaluate the effects of individual differences of fingers on rehabilitation performance using actuators, evaluation experiments were designed to measure joint ROM and joint torque on dummy fingers of different sizes. Both the modular and conventional soft actuators were used. They are characterized in Section 5.1. The evaluation experiments are presented in Section 5.2.

### 5.1. Characterizations of the Fabricated Actuators

The fabricated actuators were characterized in terms of the bending angle and tip force. Figures 5 and 6 show the experimental setup for the characterization of these two items when the modular soft actuator was set.

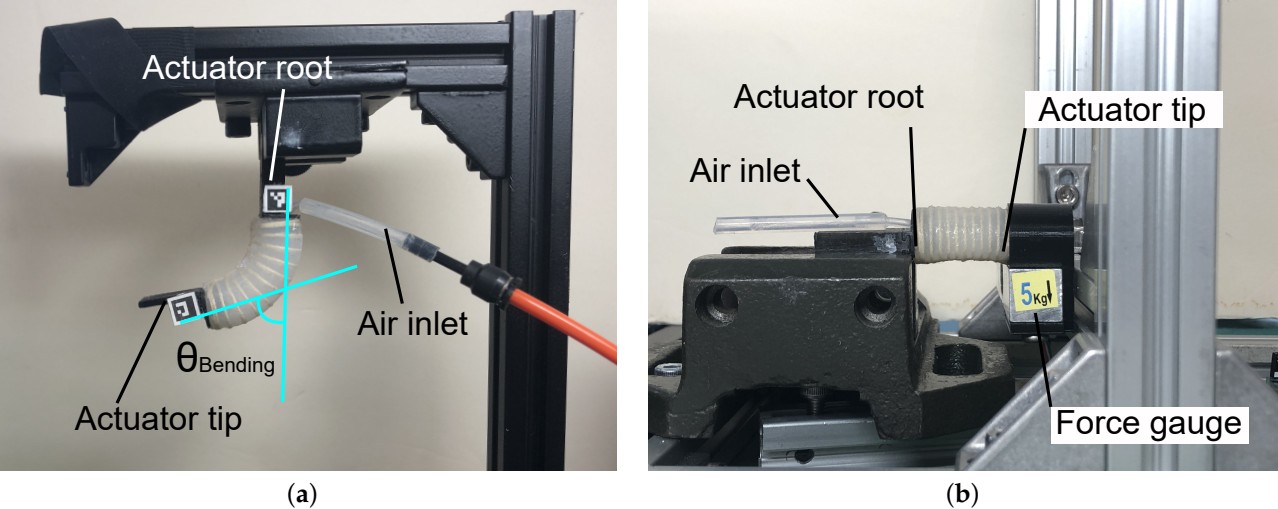

**Figure 5.** Experimental setup with only the actuator. (**a**) Bending angle; (**b**) tip force.

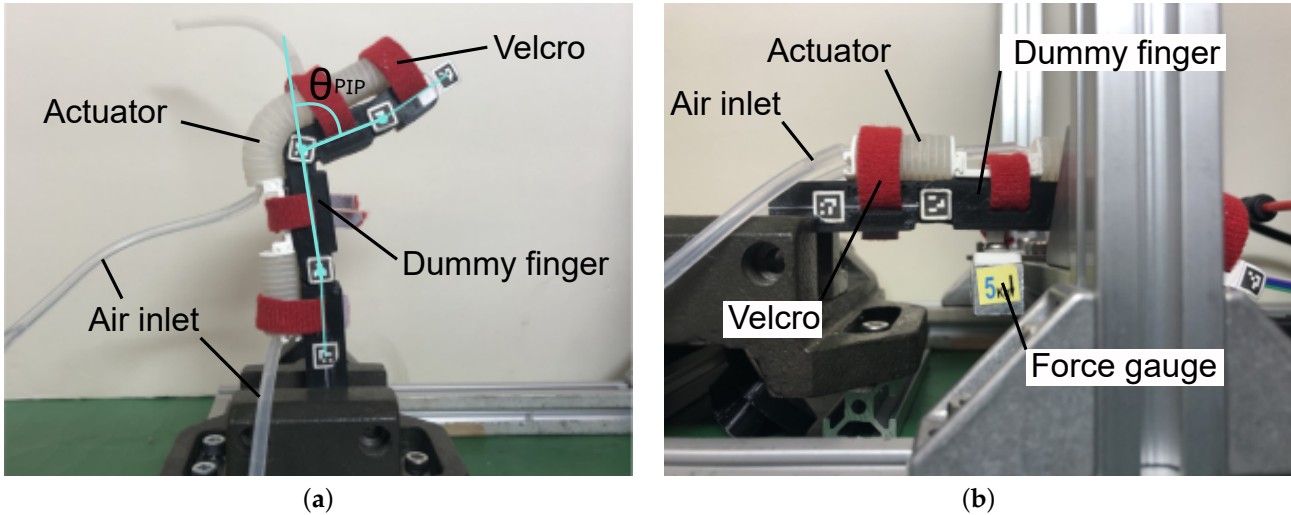

**Figure 6.** Experimental setup with actuator and dummy finger. (**a**) Joint ROM; (**b**) joint torque.

### 5.1.1. Bending Angle

The bending angle was measured to evaluate the bending performance when each actuator was pneumatically pressurized. In this measurement, the bending angle was defined as the angle between the normal vector of the actuator's tip surface and root surface (Figure 6a). The vectors were measured by a 2D motion capture system consisting of one camera and 2D markers installed at the tip and root of the actuator. The chambers at the same position of the modular and conventional soft actuators were selected and pressurized for a pair comparison. This is due to the overall structural differences between both types of actuators, as it was expected that the operation and results would differ depending on the chamber that was pressurized. The measurement environment was set such that no force other than gravity acts on the actuator. The pressure range was set to 0 to 200 kPa in all experiments.

### 5.1.2. Tip Force

The tip force was measured to evaluate how well the actuator deformation can be transmitted as the force in the desired direction. The tip force was measured with one end of the actuator fixed by a vise, and the other end was attached to a force gauge (Figure 6a). Also, for a fair comparison between the modular and conventional soft actuators, chambers at the same position of them were selected and pressurized.

### 5.2. The Evaluation Experiments

The joint ROM measurement was performed to evaluate the bending performance of each actuator on a dummy finger under more practical conditions. The joint ROM was measured by fixing each actuator to the dummy finger and tracking each joint position with a 2D motion capture system. Markers were placed at the root of the dummy finger, the MCP joint, the PIP joint, the DIP joint, and the fingertip for a total of five locations (Figure 6a). The angle of each joint was defined by the marker on the joint and the two markers above and below it. The actuators were banded in the same position and using the same Velcro for a fair comparison of the actuators.

The joint torque measurement was performed to evaluate the axial output of the joint under practical use conditions. The torque was defined as the product of the force generated during pressurization and the distance between the finger joint and the force gauge (Figure 6b). The torque of each joint was measured by fixing one side of the joint in a vise and placing the other side on the force gauge. As with the measurement of joint ROM, the actuators were fixed in the same positions and using the same Velcro straps.

In order to evaluate the possibility of individual adaptation, we measured joint ROM and torque with several dummy fingers with different sizes and joint stiffness. The measurements with dummy fingers of different sizes were taken to investigate how small changes in size affect the actuator's performance. The three types of dummy fingers (small, medium, and large) mentioned in Section 4 were used. In addition, three types of modular and conventional soft actuators (small, medium, and large) designed and fabricated for each dummy finger were used for comparison. With these dummy fingers and actuators, the cases for both when the size of the user's finger and the actuator's match and for when they do not match were simulated experimentally. The joint ROM and joint torque were measured for each size and each joint in the experiment. With this, the total number of measurements was 54, taking into consideration three types of dummy fingers, three types of joints, three actuator sizes, and two types of soft actuators.

Moreover, medium-sized dummy fingers with different joint stiffness values at the MP joint were used to investigate how the soft actuators could deal with the changes in pathological conditions such as joint stiffness. The stiffness of the MP joint of the dummy finger was reproduced by adding springs with low, middle and high spring constants, as described in Section 4. Additionally, only medium-sized dummy fingers and modular and conventional soft actuators were used to reduce external factors, which could affect the results. Although hypertension might happen in all joints, only the MP joint was tested in this study, because the underlying mechanism in terms of the driving joint with a certain stiffness is the same for all joints. The measurements were limited to the previously mentioned hyperextension of the MP joint, which is specific to people with paraplegia, and the ROM was measured for each stiffness and only for the MP joint. Six measurements were taken, taking into consideration three levels of joint stiffness, one type of joint, one actuator size, and two types of soft actuator.

## 6. Results

In this section, the bending performances of all the fabricated actuators when they are (1) pressurized independently and (2) pressurized while banded with a medium-size dummy finger are shown. After this, in order to compare the effect of individualized parameters of the modular type with those of conventional type, the bending performance of each actuator when banded with different sizes of dummy finger was revealed.

### 6.1. Bending Performance of the Modular and Conventional Soft Actuators

The bending angles of two types of soft actuators during the inflation and deflation process are shown in Figure 7. The bending angles of both the modular and conventional types increased monotonically with pressure. The maximal angles at 200 kPa, are 95.1° and 106.1°, for the modular type and conventional type, respectively. The actuator's inherent initial bending angles were also observed as 2.6° and 8.1° at 0 kPa, respectively.

With this, the ROM (obtained by subtracting the initial angle from the maximum bending angle) was 92.5° for the modular type and 98.0° for the conventional type. The hysteresis common to most pneumatic driven actuators was also confirmed for both actuator types in the air pressure-bending angle curves.

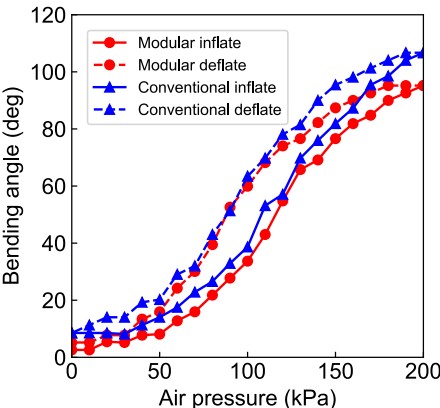

**Figure 7.** Bending angle of modular and conventional soft actuators.

The tip force of both types of soft actuator is shown in Figure 8. For both types, the tip force did not increase with pressure until 50 kPa, though a small deformation occurred. After 50 kPa, the tip force increased monotonically with pressure and achieved maximal values of 13.2 N and 6.5 N, respectively, at 200 kPa. Unstable deformation was observed in the inflation process of both types of actuators. When the actuators were inflated beyond a certain level of air pressure, the actuator's deformation turned unstable, due to the constraint given to their two ends. However, the degree of unstable deformation was larger in the conventional soft actuator than that of the modular one, which can be attributed to the elastic material between the chambers. In case of the modular actuator, the rigid connectors could prevent such deformation to some extent.

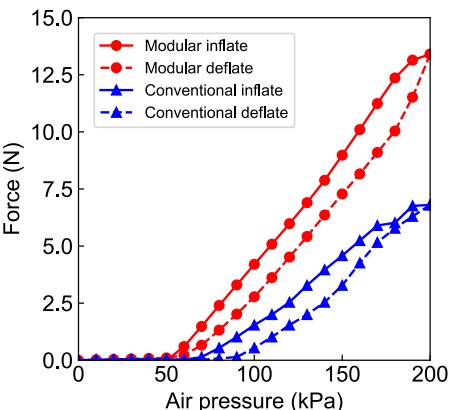

**Figure 8.** Tip force of modular and conventional soft actuators.

Moreover, even in the constrained measurement condition, hysteresis was clearly observed for both types of soft actuators, though the hysteresis is a reversed one, in which the force during deflation is smaller than that during inflation.

Finally, the results of joint ROM and torque for each joint when each type of soft actuator was mounted on the dummy finger of medium size are shown in Figures 9 and 10. The joint ROMs of both types reached their maximal values at 200 kPa. The maximal ROMs of the modular type were 80.9°, 99.6° and 89.4° for DIP, PIP, MCP joint, respectively. On the other hand, The maximal ROMs of conventional type were 66.8°, 74.6° and 69.4° for DIP, PIP, and MCP joints, respectively. In addition, the air pressure-bending angle curves of both types of soft actuators showed hysteresis.

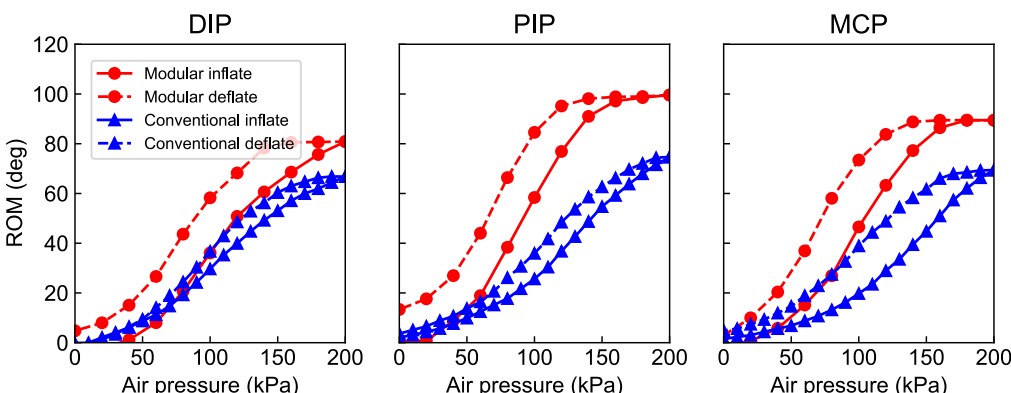

**Figure 9.** Joint ROMs when modular and conventional soft actuators were mounted to medium size dummy finger.

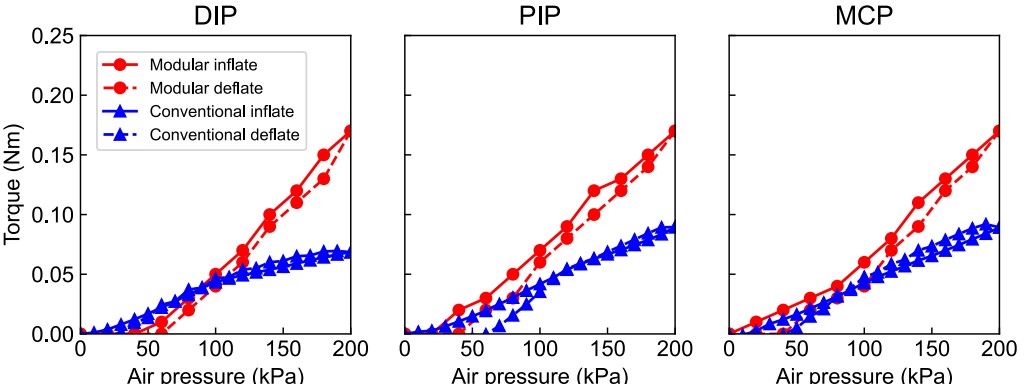

**Figure 10.** Joint torques when modular and conventional soft actuators were mounted on medium-sized dummy finger.

The joint torque of both the modular and conventional types showed the maximal values at 200 kPa. The maximal torque values of the modular type were 0.17 Nm, 0.17 Nm, and 0.17 Nm for the DIP, PIP, and MCP joints, respectively. On the other hand, The maximal torque values of the conventional type were 0.07 Nm, 0.09 Nm, and 0.09 Nm for the DIP, PIP, and MCP joints, respectively. However, the effects of hysteresis in the air pressure–torque curves were unclear. In particular, the conventional soft actuator showed similar trajectories during both the deflation and inflation processes.

### 6.2. Bending Performance of Different Actuators Banded with Dummy Fingers of Different Sizes

The joint ROM and torque when the modular and conventional actuators were banded with dummy fingers that mimic human fingers with different sizes are shown in Figures 11 and 12, respectively. The horizontal axis labels (S, M, and L) stand for the size of the dummy fingers introduced in Section 4, and each line shows the maximal ROM (Figure 11) or torque (Figure 12) of actuators of different sizes at 200 kPa. The maximal values changed with the size of both actuators and dummy fingers. The common trend from all the results is that the modular actuators always provided higher joint angle or joint torque than the conventional ones. Moreover, when the actuator and dummy finger sizes matched (e.g., actuator size M and dummy finger size M, M-M pair below), a higher ROM and torque could be achieved compared to the mismatched pairs (for the actuator size M, M-S, and M-L). The ratio of the ROM or torque of matched pair and mismatched pair reflects the significance of the size matching. Tables 2 and 3 listed $R_{ROM}$ and $R_{torque}$ (the ratio values of bending performance of matched and mismatched pairs for different joints). The $R_{ROM}$ and $R_{torque}$ were calculated as the quotient between 2× value of the matched

pair and the sum of the values of the two mismatched pairs. For example, $R_{ROM\_MM}$ and $R_{torque\_MM}$ for M-M pair are as follows:

$$R_{ROM\_MM}, R_{torque\_MM} = \frac{2 \times V_{M-M}}{V_{M-S} + V_{M-L}} \quad (5)$$

where $V_{MM}$ is a ROM or torque value of the matched pair, and $V_{M-S}$ and $V_{M-S}$ are values of the unmatched pair. If these ratios are greater than 1, it indicates that the matched pair achieved higher ROM or torque than the mismatched pairs. For both modular and whole-finger actuators, the difference in ROM between different actuator size (small, middle, large) is smaller for MCP than for the other joints. This is because of the relative position of the actuator and dummy finger in the experiment. For all the matched or mismatched pairs, the root of the actuators is always aligned with the MCP joint, which makes the effect of mismatch in MCP smaller than that of the other two joints. The torque measurements also showed different chamber deformations for matched and unmatched pairs in the pneumatic pressurization process (Figure 13).

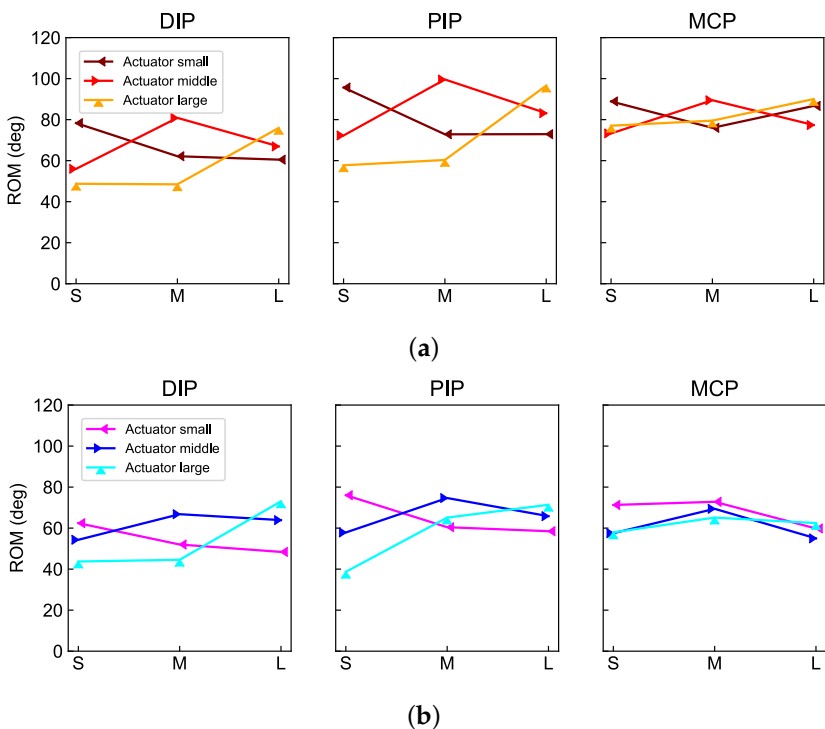

**Figure 11.** Maximal joint ROMs at 200 kPa for different size dummy finger. (**a**) Modular type; (**b**) conventional type.

**Table 2.** $R_{ROM}$ (the ratio of the ROMs of matched pair and mismatched pair).

|  | DIP Joint | | | PIP Joint | | | MCP Joint | | |
|---|---|---|---|---|---|---|---|---|---|
|  | **SS** | **MM** | **LL** | **SS** | **MM** | **LL** | **SS** | **MM** | **LL** |
| Modular | 1.5 | 1.5 | 1.2 | 1.5 | 1.5 | 1.2 | 1.2 | 1.2 | 1.1 |
| Conventional | 1.3 | 1.4 | 1.3 | 1.6 | 1.2 | 1.1 | 1.2 | 1.0 | 1.0 |

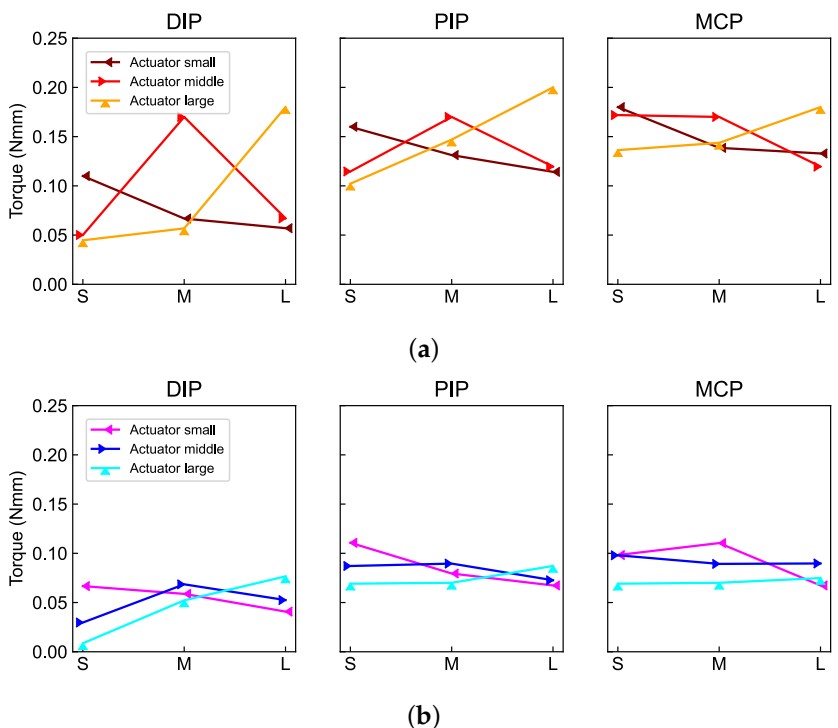

**Figure 12.** Maximal torque at 200 kPa for different size dummy finger. (**a**) Modular type; (**b**) conventional type.

**Table 3.** $R_{torque}$ (the ratio of the torques of matched pair and mismatched pair).

| | DIP Joint | | | PIP Joint | | | MCP Joint | | |
|---|---|---|---|---|---|---|---|---|---|
| | **SS** | **MM** | **LL** | **SS** | **MM** | **LL** | **SS** | **MM** | **LL** |
| Modular | 2.3 | 2.8 | 2.9 | 1.5 | 1.2 | 1.7 | 1.2 | 1.2 | 1.4 |
| Conventional | 3.5 | 1.2 | 1.6 | 1.4 | 1.2 | 1.2 | 1.2 | 1.0 | 1.0 |

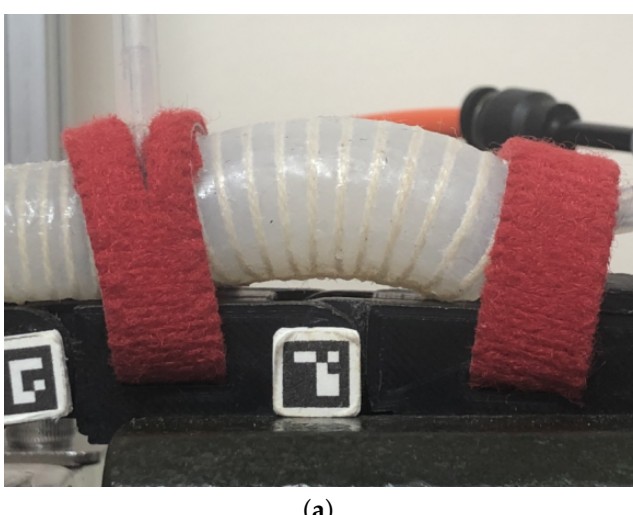 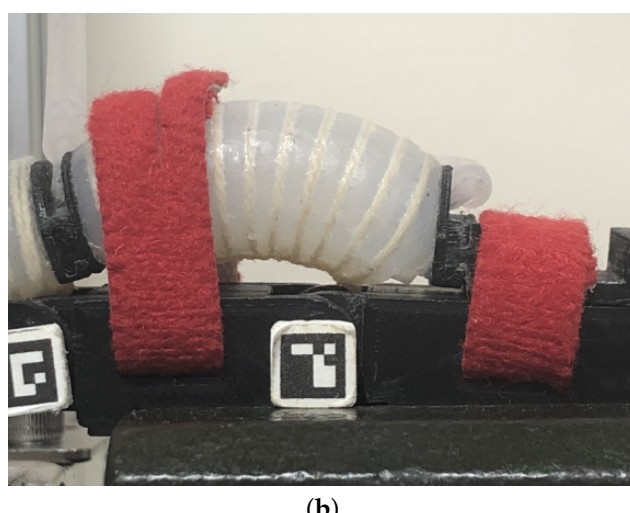

(**a**)                                                              (**b**)

**Figure 13.** Different chamber deformations for matched and unmatched pairs. (**a**) Matched pair M-M; (**b**) unmatched pair M-L.

The results of the joint ROM of dummy fingers with different joint stiffness levels (no spring, low, mid, and high) are shown in Figure 14. Both the actuator and dummy finger are of medium size. The ROM values shown are the maximal angle obtained in the dummy finger MCP joint installed with springs for different stiffness levels. These results show

that the joint ROM decreases as the joint stiffness increases for both types. The modular actuator provided a wider ROM for all stiffness levels, although as the spring constant increases, it decreases faster than the conventional one (Table 4). The rate of change of a certain level of stiffness was calculated as the ratio of ROM resulting from the stiffness, and that of the no spring case.

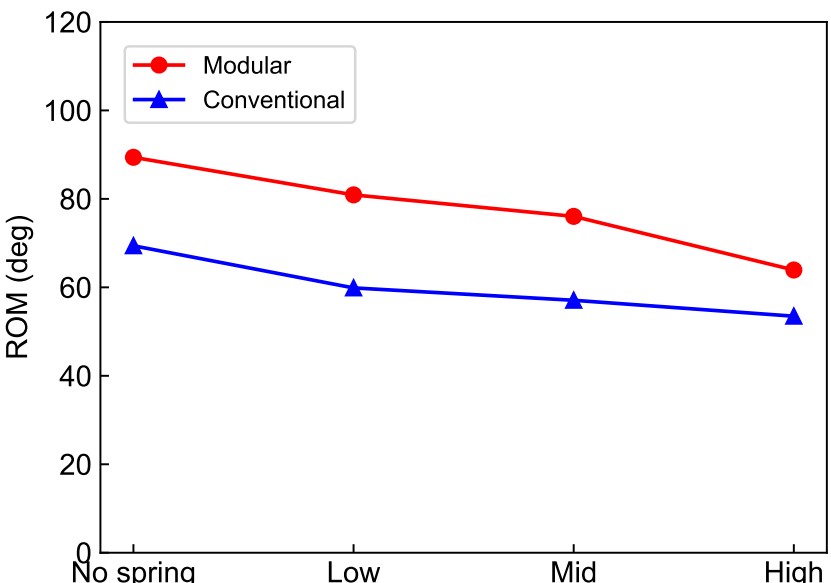

**Figure 14.** Maximal joint ROMs at 200 kPa for different joint stiffness values.

**Table 4.** Maximal ROM value and rate of change for different joint stiffness.

|  | No Spring | Low | Mid | High |
|---|---|---|---|---|
| Modular (deg, %) | 89.4, N/A | 80.9, −9.4 | 76.0, −14.9 | 63.9, −28.4 |
| Conventional (deg, %) | 69.4, N/A | 59.9, −13.7 | 57.1, −17.7 | 53.5, −22.9 |

## 7. Discussion

### 7.1. Comparison of the Bending Performance of Two Types of Soft Actuators

Both the modular and conventional soft actuators showed similar bending angles at 200 kPa (due to monotonicity, maximal bending angles among the range of tested pressure: values 92.5° vs. 98.0°). This may be due to both types of soft actuators having the same chamber size. Chamber dimensions have an effect on an actuator's deformation, and thus on its bending performance. In the case of the modular actuator, the rigid connectors between each module limit the deformation at both ends of the actuators during pressurization.

On the other hand, the modular soft actuator's maximal tip force (the tip force at 200 kPa) is much higher than that of the conventional one (13.2 N vs. 6.5 N). This may be because of the rigid connectors of the modular soft actuator. In the conventional soft actuator, the elastic material between two successive chambers may absorb the force generated during measurement as the elastic material can be easily deformed by air pressure and by the constraints for mounting the actuator on the finger. However, in the modular soft actuator, the force generated can be transmitted more efficiently through the rigid connectors, which allow for a bigger tip force.

Observing these results, it is reasonable to declare that the modular design has better overall bending performance (angle and force), which agrees with the findings reported by related studies [16,29].

When the actuators were banded to the dummy finger (medium size, no spring at MCP joint), the ROMs of the joints of the dummy finger with the modular soft actuator at 200 kPa

were much higher than those of the conventional one. Compared with the measurement results of actuators alone, the difference in bending angles achieved by the modular and conventional soft actuator is much bigger when banded with the dummy fingers. This might be due to the fact that the conventional soft actuator could not overcome the resistance caused by the dummy finger's joint structure and by the stiffness of each joint. In fact, the torque generated in the joints of the dummy finger by the two types of soft actuators provided evidence for this hypothesis: the torque generated by the modular type is much bigger than that of the conventional type. Moreover, the ROMs achieved by the modular soft actuator are quite close to those of healthy subjects [31]. The hysteresis observed in the bending angle–air pressure and torque–air pressure curves could be interpreted as separate phenomena. The hysteresis of the bending angle–air pressure curve could have been due to the non-linear behavior of elastic materials. This phenomenon has also been reported in related studies [29,30]. On the other hand, the hysteresis of the torque–air pressure curve is smaller than that of the bending angle–air pressure curve, and the transition is also reversed. This might be because the non-linearity was weakened by fixing a part of the deformation of the actuator during measurement, and the order of deformation was reversed. In summary, the joint ROM and joint torque results showed that the proposed modular actuator is appropriate for hand finger rehabilitation.

### 7.2. Importance of Size Matching between Actuators and Dummy Fingers

The results of the bending angle and torque of matched and mismatched actuator-dummy finger pairs showed that: (1) the modular soft actuators of all matched pairs, not only the M-M pair but also S-S and L-L pairs, have a better bending performance (e.g., ROM, torque) than their conventional counterparts; (2) the bending performance of matched pairs is much better than that of mismatched pairs of both the modular and conventional soft actuators; (3) the effect of matching can be disclosed by $R_{ROM}$ and $R_{torque}$ (the ratio of bending performance of matched and mismatched pairs)—higher $R_{ROM}$ and $R_{torque}$ values means the bigger influence of mismatching, and thus, the higher necessity of size matching. As shown in Figure 11, the modular soft actuators resulted in a higher $R_{ROM}$ and $R_{torque}$ value than the conventional ones, and for both types, the $R_{ROM}$ and $R_{torque}$ values of DIP, PIP and MCP joints are presented a descending order. It is reasonable to claim that size mismatching is a common problem for both types of soft actuators with these results. However, the mismatching of modular soft actuators and dummy fingers could be easily resolved by adjusting the size of rigid connectors.

In order to further understand why mismatched pairs could result in worse bending performance and, even worse, even risk joint injury, a set of equations of equilibrium were employed to analyze the interaction between actuators and dummy fingers for matched and mismatched pairs.

Figure 15 shows a free body diagram (FBD) of matched and mismatched pairs. For simplicity, only one joint and the chamber for its bending were modeled. However, this same analysis could be applied to multi-joint cases. As shown in the figure, the length from the joint center to the force measurement point is $r$ = 15 mm. The load cell was set up such that it is perpendicular to the bottom plane of the dummy finger. The force equilibrium equations for the force pressing the load cell by dummy finger and the torque at the joint are as follows:

$$R_l = F_d \tag{6}$$

$$\tau = rF_d \tag{7}$$

where $R_l$ is the reaction force from the load cell and $F_d$ is the force pressing the load cell by the dummy finger. As shown in Figure 15a, in the case of matched pairs, because the sizes of the dummy finger and actuator are well matched, there is no change from the unpressurized set position in the direction of the actuator's force pressing the dummy finger when the chamber is inflated. Moreover, the force equilibrium equations for the

actuator's pressing force to the dummy finger and torque generated for the matched pair are as follows:

$$R_d = F_{ac} \tag{8}$$

$$\tau = rF_{ac} \tag{9}$$

where $R_d$ is the reaction force from the dummy finger and $F_{ac}$ is the force applied to the dummy finger by the actuator. Because $R_d$ and $R_l$ are equivalent, from Equations (6)–(9), it is clear that the force of the actuator generates the torque in this case.

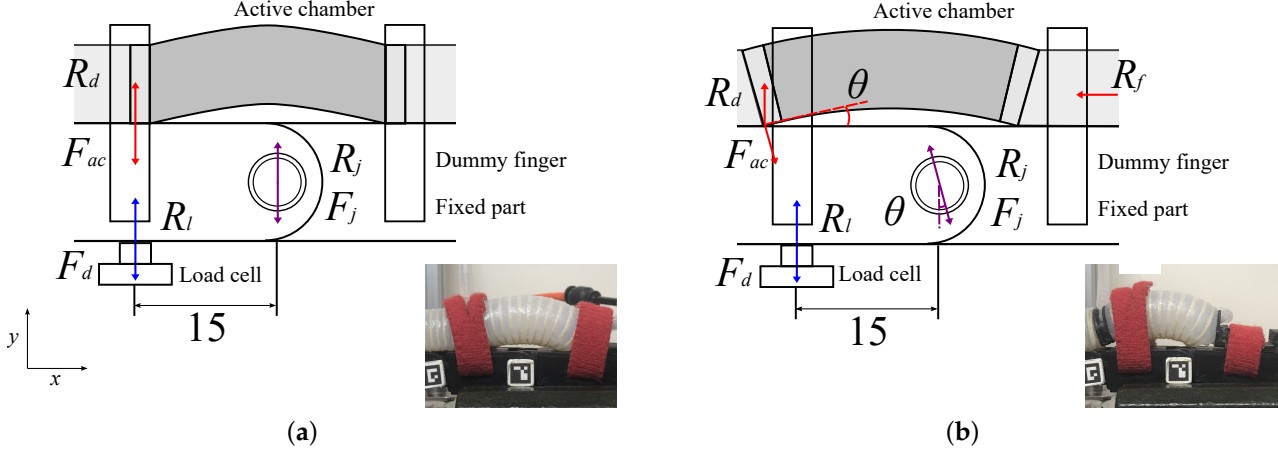

**Figure 15.** Free body diagrams and experimental setting. (**a**) Right size selected; (**b**) different size selected.

Figure 15b shows the FBD of the mismatched pair. The mismatch induces a displacement between the joint and the chamber position, which results in a deformation of the chamber edge, and thereby the angle $\theta$ at which the actuator presses the dummy finger, changing from the unpressurized set position. The force equilibrium equations of the actuator and dummy finger in the mismatched pair are as follows:

$$R_d = F_{ac} \cos \theta \tag{10}$$

$$\tau = rF_{ac} \cos \theta \tag{11}$$

The torque generated by the actuator to the dummy finger is shown as Equation (11). It is clear that only the portion of force that is perpendicular to the dummy finger contributed to the torque in this case. Thus, Equations (10) and (11) disclose the influence of actuator–dummy finger mismatching. These support the experimental results that showed a matched pair resulted in larger torque than mismatched pairs.

The analysis based on force equilibrium can also predict the load onto the finger joints. As shown in Figure 15a, the force acting on the joint $F_j$ has the same magnitude and direction as $F_{ac}$, which is the force generated by the actuator. The $F_{ac}$ can be decomposed to $F_{acy}$, which works to rotate the joint, and $F_{acx}$, which does not contribute to the rotation (Equation (12)), but may cause an extra burden to finger joints.

$$\begin{aligned} x : F_{acx} &= F_{ac} \sin \theta \\ y : F_{acy} &= F_{ac} \cos \theta \end{aligned} \tag{12}$$

The magnitude of $F_{acx}$ depends on the angle $\theta$ of the force applied to the finger by the actuator. A large deviation in size between the actuator and the finger leads to an increase in $F_{acx}$. For example, in the experiment with a mismatched actuator and dummy finger pair, the angle $\theta$ was around $10°$. Since the tip force of the actuator $F_{ac}$ confirmed in the experiment with the modular soft actuator alone was 13.2 N at 200 kPa, $F_{acx}$ could be estimated to be 2.3 N by the $\theta$, $F_{ac}$ and Equation (12). In a simulation study, Butz et al.

estimated the joint force with a finite element model and reported that the joint force during typing in daily life was 2.5 N for DIP joints [34]. Although $F_{acx}$ is not big, as a long-term load on joints, it might lead to unexpected injuries [34]. Thus, matching the actuator with finger is important for efficient and safe function assistance.

It should be noted that the deformation of the chamber edge is not the only factor that causes an excessive load to the joint. The misalignment between the axis of rotation of the actuator and that of the joint, as well as the abnormal concentration of pressure due to the misalignment can be the factors, too. Though, the deformation of the chamber edge has been clearly observed in our experiments, and analyzed as one major factor. Further investigation is necessary.

Moreover, especially for DIP joints, the modular soft actuators resulted in a much higher torque difference between mismatched and matched pairs (2.7 vs. 2.1, $R_{torque}$ averages in DIP joints) than that of conventional ones. For example, when a small actuator is mounted onto a finger, the actuator does not cover the fingertip, which reduces the contact area, and thus torque generated to the DIP joint. Similarly, when a large actuator is mounted to the finger, chamber edge where the contact force generates goes beyond the fingertip, resulting in a smaller contact area. Another reason for this is that it is difficult to band the 3D-printed rigid part, which is attached to the DIP for adjusting to the finger length, onto the fingertip with Velcro; thus, a mismatched soft actuator could deliver smaller torque to dummy fingers compared to a matched pair.

Thus, it is reasonable to deduce that, although compared to an assistive device with link mechanisms, soft actuators do not require precise adjustments of the center of rotation; they need to match the finger size of the individuals to be assisted, or the expected performance might not be achieved, and safety might not be guaranteed.

When tested with dummy fingers with different joint stiffness values, modular soft actuators could achieve higher maximal ROM than conventional ones, and the bending angle decreased as the joint stiffness increased. Apparently, the modular soft actuators could deliver force better than the conventional ones. This is due to the use of rigid connectors that have an efficient force transmission. Other actuator design factors (e.g., chamber geometry and reinforcement fiber design) were common to the modular and conventional types, so the contribution of rigid connectors to force transmission is likely to be significant.

In addition, the modular soft actuator could achieve ROM over 70° for the dummy finger joint with middle-level (mid) stiffness. Yamada et al. confirmed with a rehabilitation physician that a flexion of around 70° in the MP joint is sufficient to support hand rehabilitation [35]. Therefore, the results showed that the proposed modular soft actuator could accommodate a range of joint stiffness values for dealing with the individual differences in these biomechanical characteristics.

In this study, experiments were conducted with three different sizes of two types of soft actuators, and three sizes of dummy fingers designed based on the index finger length and joint stiffness data from Human Hand Dimensions Data for Ergonomic Design 2010 [24]. These experiments highlighted the following two points: (1) the importance of size matching between soft actuators and individual finger size, and thus, the necessity to take into consideration the individual differences during the design of soft rehabilitation glove was shown; (2) it was made clear that modular actuators are preferable for dealing with stiffness disorder of specific joints, and accommodating soft actuators for individual fingers.

However, there are some limitations of the current study. Firstly, only three levels of finger size and stiffness might not be able to give generalized findings. It is necessary to perform an examination with finer levels of finger size changes and joint stiffness values. Secondly, the current dummy finger structure is simple and optimized for measurement but different from that of human fingers. Therefore, it is necessary to design a dummy finger with realistic anatomical details to further investigate individual differences' effects. Finally, more theoretical and mathematical analysis is also needed to facilitate the extrapolation of

this work to a wider range of soft actuators. Therefore, developing FEA (Finite Element Analysis) models of finger-soft actuator complexes and analyzing the interaction between them, and the effect of the interaction is important future work for us.

## 8. Conclusions

The purpose of this study is twofold: it proves the advantage of fiber-reinforced modular soft actuators developed as an assistive and rehabilitation device over the conventional ones, and it assesses the effects of individual differences on finger motion assist and rehabilitation. So far, the effects of individual differences in biomechanical characteristics on hand assist and rehabilitation have not been investigated in depth. Moreover, the individualization of soft actuators as rehabilitation tools has not been focused on. Thus, concerns about the potential safety of rehabilitation assisted with soft actuators have been left unaddressed. In this study, fiber-reinforced modular soft actuators were developed and compared with conventional ones, in terms of bending performance of actuators independently, and with the actuators mounted on dummy fingers that reproduced individual differences in finger size and MP joint hyperextension, which is often seen in stroke patients. As a result of comparison and evaluation, it was confirmed that modular soft actuators have better bending performance, and the bending performance of mismatched pairs of dummy fingers and actuators is worse than that of matched pairs. Moreover, modular soft actuators could deal with the different joint stiffness values better than conventional ones. Since our fiber-reinforced modular soft actuators design strategy and approach to assess the adaptability from individual differences can be extended to all SRG research in the future, this study pushes soft actuators one step closer to practical use for efficient and safe robot-assisted rehabilitation.

**Author Contributions:** S.K. was the main author of the manuscript and conducted the design, fabrication and testing stages of the presented prototypes. Y.W., P.E.T.V., Y.L., S.H., R.N. and Y.-H.H. assisted in data analysis and their presentation and participated in revising the manuscript's contents. W.Y. was the main supervisor and advisor of the work and contributed to the manuscript by reviewing and revising its contents. All authors have read and agreed to the published version of the manuscript.

**Funding:** This research received no external funding.

**Institutional Review Board Statement:** Not applicable.

**Acknowledgments:** We would like to thank Zhongchao Zhou for technical support in the project. We are also grateful to Xinyao Guo for providing valuable comments on the contents of the manuscript. Finally, we would also like to thank Kornkanok Tripanpitak for providing invaluable insight about physical therapy.

**Conflicts of Interest:** The authors declare no conflict of interest. The funding sponsors had no role in the design of the study; in the collection, analyses or interpretation of data; in the writing of the manuscript; nor in the decision to publish the results.

## Abbreviations

The following abbreviations are used in this manuscript:

| | |
|---|---|
| SRG | Soft rehabilitation glove |
| ROM | Range of motion |
| DP | Distal phalanx |
| PP | Proximal phalanx |
| MP | Medial phalanx |
| DIP | Distal interphalangeal (joint) |
| PIP | Proximal interphalangeal (joint) |
| MCP | Metacarpophalangeal (joint) |
| FBD | Free body diagram |

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
