# Peer review of "Evaluation of Fiber-Reinforced Modular Soft Actuators for Individualized Soft Rehabilitation Gloves"

_actuators, doi:10.3390/act11030084_

Round 1

Reviewer 1 Report

The work presented is important.  A better understanding of soft actuators and how they can be best used for rehabilitative and/or assistive robotic devices has the potential to significantly impact the field. The idea of using a modular design is worth exploring, as is the use of reinforcement with fiber or otherwise.

My main concern is the impact of this paper.  It has similar content and structure to one of the authors' previous works, ref [10], also published in Actuators.  As presented, it will have minimal impact, only making a significant difference to those studying very similar actuators.  Essentially, the authors have tried a couple of new things (modular, fiber reinforced) and tested those ideas, but the work lacks mathematical or engineer analysis or techniques to facilitate extrapolating this work to a broader range of soft actuators.  A mathematical model and/or FEA approach validate by experimental results would significantly improve the impact of this work.

Author Response

添付ファイルをご覧ください。

Reviewer 3 Report

This work aimed to clarify the effects of individual differences on the actuator’s performance through a series of experiments using dummy fingers designed with individualized parameters. Two types of fibre-reinforced soft actuators were developed and compared: modular type for each joint and conventional (whole-finger assist) type. It was found that the modular soft actuators respond better to individual differences set in the experiment and exhibit superior performance than the conventional ones. By suitable connectors and air pressure, the modular soft actuators could cope with the individual differences with minimal effort. The effects of the individualized parameters are discussed, and design considerations are extracted and summarized. This study will play an important role in pushing the SRGs to actual rehabilitation practice.

However, I have a big concern.

The research is exciting and valuable, except that the paper emphasizes that the different degrees of softness and hardness for individual differences and the effect on the patient's recovery process can be very different. But throughout the paper, there is nothing about the test with people's fingers, a rehabilitation training test experiment, and the user feedback effect. The whole paper has only simple comparative data and does not really verify the real effect through actual tests on users. I think this aspect is essential.

Round 2

Reviewer 1 Report

The authors have done some good work address some concerns from all reviewers, but I still think the paper has insufficient theoretical and mathematical analysis which is limiting the impact of the work.

Reviewer 3 Report

The authors answered all my questions very well and I recommend receiving this paper.
